# Design and Optimization for Mounting Primary Mirror with Reduced Sensitivity to Temperature Change in an Aerial Optoelectronic Sensor

**DOI:** 10.3390/s21237993

**Published:** 2021-11-30

**Authors:** Meijun Zhang, Qipeng Lu, Haonan Tian, Dejiang Wang, Cheng Chen, Xin Wang

**Affiliations:** 1Key Laboratory of Airborne Optical Imaging and Measurement, Changchun Institute of Optics, Fine Mechanics and Physics, Chinese Academy of Sciences, Changchun 130033, China; zhangmeijun@ciomp.ac.cn (M.Z.); tianhaonan@ciomp.ac.cn (H.T.); wangdj04@ciomp.ac.cn (D.W.); chengengeng@126.com (C.C.); 2University of the Chinese Academy of Sciences, Beijing 100049, China; 3School of Mechanical and Aerospace Engineering, Jilin University, Changchun 130025, China; wxin@jlu.edu.cn

**Keywords:** primary mirror, sensitivity analysis, optimization design, modulation transfer function, thermal deformation resistance

## Abstract

In order to improve the image quality of the aerial optoelectronic sensor over a wide range of temperature changes, high thermal adaptability of the primary mirror as the critical components is considered. Integrated optomechanical analysis and optimization for mounting primary mirrors are carried out. The mirror surface shape error caused by uniform temperature decrease was treated as the objective function, and the fundamental frequency of the mirror assembly and the surface shape error caused by gravity parallel or vertical to the optical axis are taken as the constraints. A detailed size optimization is conducted to optimize its dimension parameters. Sensitivities of the optical system performance with respect to the size parameters are further evaluated. The configuration of the primary mirror and the flexure are obtained. The simulated optimization results show that the size parameters differently affect the optical performance and which factors are the key. The mirror surface shape error under 30 °C uniform temperature decrease effectively decreased from 26.5 nm to 11.6 nm, despite the weight of the primary mirror assembly increases by 0.3 kg. Compared to the initial design, the value of the system’s modulation transfer function (0° field angle) is improved from 0.15 to 0.21. Namely, the optical performance of the camera under thermal load has been enhanced and thermal adaptability of the primary mirror has been obviously reinforced after optimization. Based on the optimized results, a prototype of the primary mirror assembly is manufactured and assembled. A ground thermal test was conducted to verify difference in imaging quality at room and low temperature, respectively. The image quality of the camera meets the requirements of the index despite degrading.

## 1. Introduction

Aerial optoelectronic sensors have the advantages of flexibility, high resolution and strong timeliness, which are widely used in the fields of topographic survey, emergency rescue, and national defense security, etc. [1,2,3]. Structural design of an aerial optoelectronic sensor is not only subject to strong constraints of volume and weight. Furthermore, it needs to have good environmental adaptability, be capable of withstanding a wide range of temperature change (−40~+55 °C), shock, vibration, overload and other mechanical environments impact, so as to output clear images. The primary mirror, as the critical component of a high-resolution optoelectronic sensor, is sensitive to environmental factors, such as temperature variation and external acceleration. How to reduce the sensitivity of the supporting structure of the primary mirror to ambient temperature change becomes an issue worthy of serious consideration. A stable and reliable support structure plays an important role in ensuring image quality.

There has been a lot of research on design and optimization for mounting the mirrors. The parameter optimization or topology optimization method is widely used to optimize the structural parameter on a predefined configuration [4,5,6,7]. Shao Mengqi et al. [8] established a size parameter optimization model to optimize the optical quality of the mirror. Contribution of size parameters to optical performance was further evaluated with the help of ray-tracing, multidisciplinary optimization and finite element analysis software. Kihm Hagyong et al. [9,10] proposed a design method to optimize a primary mirror and its flexure separately in a space telescope. A multi-objective genetic algorithm was implemented to optimize the design parameters which met the design goals and reduced the optimization time by an order of magnitude. Liu Shutian et al. [11] presented a design process of primary mirror based on topology optimization, in which structural compliance was treated as the objective function to minimize the optical surface deviation due to self-weight and polishing pressure loading. According to the simulation result, the configuration of a lightweight mirror for a large-aperture space telescope was obtained. Hu Rui et al. [12] developed a method for designing the flexure for mounting the primary mirror. A topology optimization was firstly adopted to the extraction of conceptual configuration. Then a detailed shape optimization was conducted to optimize the dimension parameters. To reduce the sensitivity of a lightweight mirror to the mount location, the eccentricity of the hyperbola curve was introduced as the objective function in the topology and parameter optimization by Jiang Ping et al. [13]. An optimal lightweight design for a zerodur primary mirror with an outer diameter of 566 mm and supporting bipod flexure was studied by Chen YIcheng et al. [14], in which the lightweight ratio and the surface shape error for the deformed surface under polishing pressure and gravity was achieved. 

At present, there are few studies on the optimization of mirrors and their supporting structures together in aerial photoelectric sensors, despite most of this research on the optimization for lightweight mirrors and its supporting structures being focused on space-borne or ground-based telescope applications. Due to the difference in the working environment of the aviation photoelectric remote sensor and the aerospace remote sensor, the performance requirements are different. The operating temperature of the internal optical system of the aerospace remote sensor can usually be maintained within ±5 °C through the internal thermal control facility. However, the working temperature of the optical system inside the aerial remote sensor can usually be maintained within −40~+55 °C with poor thermal control measures or none. Meanwhile, the design or optimization process for supporting the mirror is usually separated into two independent parts: mirror design and flexure design. In fact, the design or optimization of the mirror is coupled with the flexure design. It is difficult to obtain better optimization results by just studying the optimization of the mirror itself or its supporting structure alone.

In this paper, the optimization design of the mirror and its supporting structure is combined and studied together. High thermal deformation resistance of the primary mirror of an aerial optoelectronic sensor is considered for better adaptation in the low or high temperature working environment. A detailed size optimization of the primary mirror and the flexure is conducted together to optimize their dimension parameters. The process of optomechanical analysis and optimization for the primary mirror assembly are developed in sequence. The mirror surface shape error caused by uniform temperature decrease was treated as the objective function and the fundamental frequency (the first-order natural frequency) of the mirror assembly and the surface shape error caused by gravity parallel or vertical to the optical axis are taken as the constraints. The influence of size parameters on the mirror surface shape error is analyzed. As a result, the configuration of the primary mirror and the flexure is obtained. Compared with the initial design, the optical performance of the system has been significantly improved. 

## 2. Performance Requirements

The aerial optoelectronic sensor contains a visible light camera which adopts the Cassegrain optical system in this paper. The visible light camera is integrated in the spherical envelope space, and the focal length is 1000 mm, which is easily influenced by the thermal disturbances and external acceleration. As the core component of the optical system, the primary mirror needs to adapt to the temperature change of the environment to the greatest extent and minimize the degradation of image quality. Figure 1 shows the optical model of the camera which is established in the ray-tracing software Zemax for this study.

Usually, the optical axis of the mirror is placed horizontally on the air-floating platform, which is conducive to assembly and optical testing. In this situation, the optical performance can be adjusted to the best state, almost close to the diffraction limit. In fact, the optical system is installed on the airplane, in actual use scenarios, and most of the time it is in the working state of squint imaging. This inconsistency between alignment status and usage status will inevitably, more or less, lead to a decline in the performance of the optical system. Namely, the supporting structure of the primary mirror needs to have sufficient rigidity to resist gravity deformation or deformation caused by external acceleration [15]. Furthermore, the working temperature of the optical system inside the aerial optoelectronic sensor in this paper can usually be maintained within a wide range of temperature change (−10~+50 °C) with an active-passive combined thermal control measures. In the airborne working environment, the temperature environment has a more direct impact on the image quality of the optical system compared to other environmental factors such as gravity. Therefore, the supporting structure of the primary mirror needs to have adequate compliance to resist thermal deformation.

A convenient measure to quantify the optical performance of the mirror is RMS (root-mean-square) of the mirror surface shape error. The RMS can be obtained by Zernike polynomial fitting, which is expressed by Equation (1): (1)$$RMS=\sqrt{\sum_{i=1}^{N}w_{i}\delta_{i}^{2}},\delta_{i}=u_{i}-z_{1}-z_{2}-z_{3}$$
where *N* is the number of the nodes of the mirror face, *w_i_* is weight of the *i*-th node, *u_i_* is the sag displacement of the *i*-th node, *z*_1_, *z*_2_, *z*_3_ represent the piston, tilt and power term of the deformed surface, respectively [12].

Due to the orthogonality of the Zernike polynomials over a unit circle, they are convenient to use to describe the aberration in optical engineering [16,17]. The mathematical description of Zernike polynomials for a deformed surface are defined by Equation (2): (2)$$\Delta Z(\rho ,\theta )=A_{00}+\sum_{n=2}^{\infty }A_{n0}R_{n}^{0}(\rho )+\sum_{n=1}^{\infty }\sum_{m=1}^{n}R_{n}^{m}(\rho )\left[A_{nm}\cos(m\theta )+B_{nm}\sin(m\theta )\right]$$
where Δ*Z*(*ρ*,*θ*) denotes the deformed surface, *ρ* is the normalized radius on the reflection surface, *θ* is the azimuth angle, *A_nm_* and *B_nm_* are Zernike coefficients in *x* direction and *y* direction, the variables *n* and *m* are radial and circumferential wave number, respectively. 

The radial dependence $R_{n}^{m}(\rho )$ of the Zernike polynomials can be expressed by Equation (3):(3)$$R_{n}^{m}(\rho )=\sum_{k=0}^{(n-m)/2}\frac{{\left(-1\right)}^{k}(n-k)!}{k!\left[(n+m)/2-k\right]!\left[(n-m)/2-k\right]!}r^{\left(n-2k\right)}$$

The goal of this paper is to further optimize the thermal adaptability of the supporting structure of the primary mirror. Thus, the mirror surface shape error caused by uniform temperature decrease was treated as the objective function, and the fundamental frequency of the mirror assembly and the surface shape error caused by gravity parallel or vertical to the optical axis are taken as the constraint. The RMS of the mirror under gravity and thermal load separately should be less than *λ*/40 (where *λ* = 632.8 nm). The mass of the primary mirror component should be less than 1.6 kg including the primary mirror, invar sleeves and the flexures. The fundamental frequency of the primary mirror component should be more than 200 HZ, which indicates the stiffness to resist the gravity deformation or external acceleration. 

Some performance requirements are requested in our camera. Generally, there should be a certain safety margin in the structural design [18]. According to the decomposition of the camera specifications and applications, the performance requirements of the primary mirror assemblies are listed in Table 1. 

## 3. Optimal Design of the Primary Mirror Assembly

### 3.1. Initial Design of the Primary Mirror Assembly

Classical configurations of lightweight mirror include closed back, semi-closed back and open back. The open back configuration has the advantages of easy processing and high lightweight ratio, which is very suitable for an aerial camera [19]. Consequently, an open back configuration is employed in this article. The mirror usually consists of faceplate and ribs from behind. A common lightweight pocket shape could be triangular, round, hexagon or sector holes [20]. Due to the use of SiC material with high specific stiffness, sector holes are adopted for the purpose of further improving the lightweight ratio in this article. The ribs with certain thickness support the thin faceplate form behind, which not only improves the stiffness and cut down weight of the base, but also provides channels for heat conduction. Thermal influence form the surrounding environment can be partially weakened by this means [8]. 

A typical three-point back support is applied for the primary mirror in this study on account of its simplicity and reliability, as shown in Figure 2. The diameter of the primary mirror is 190 mm, and the diameter of the inner hole is 44 mm. The mirror surface of the primary mirror is a parabolic surface with high steepness, and the radius of curvature is 397.1 mm. The radial position of the supporting hole of the primary mirror is located at the equivalent radius of gyration. To minimize the mirror surface distortion when the optical axis of the mirror is no longer vertical, the axial position of the supporting connecting surface of the primary mirror is approximately located near the center of gravity plane [20,21]. The primary mirror assembly includes the primary mirror, the invar sleeves and the flexures. The mass of the primary mirror assembly is 1.3 kg and the lightweight ratio is 45.8% for the initial lightweight design. The three invar sleeves are separately bonded into the support holes of the mirror, and the three flexures are also respectively fastened on the sleeves with screws. The primary mirror is indirectly supported by three parallel flexures through support holes at the back of the mirror. The invar sleeves and three flexures are circularly symmetrical around the optical axis. The chosen material of the flexure is TC4, which has a high specific stiffness. The selected material of the sleeve is artificially prepared, which has a similar coefficient of thermal expansion (CTE) to SiC. The chosen material properties of the primary mirror assembly are shown in Table 2.

Flexure elements are usually adopted to produce elastic deformation, so as to absorb the strain energy caused by mounting and thermal affect which are a convenient means to mount a mirror with high-performance requirements. Typical flexures used consist of a notch hinge, cross hinge or leaf type. A rotationally symmetric leaf hinge provides compliance in defined axes for the purpose of precise motion control, which is easily formed from cylindrical tubes or circular disks by electrical discharge wire-cutting. This type of flexure provides three freedom, one translation and two rotations [22,23,24].

### 3.2. Establishment of Finite Element Model (FEM) 

FEA is employed to determine the deformation of the primary mirror in the following scenarios: (1) deformation in the case of 30 °C uniform temperature decrease; (2) self-weight deformation when the optical axis of the mirror is horizontal or vertical. The establishment of the finite element model is put forward through the finite element pre-processing software Hypermesh. Shell elements are used for mesh generation, with a total number of 45,784 elements and 46,585 nodes. Rigid elements are established to simulate glue and screw connections. The mirror surface shape error RMS (removal of piston, tilt and power) is converted into a response equation in DRESP2 format to perform Zernike polynomial fitting of the deformed mirror surface, and then imported into Optistruct solver for size optimization analysis. The design variables involved in this article are divided into two types: gauge variables and shape variables. Gauge variables are particular cases of size variables, where the design variables are two-dimensional properties. The shape optimization technique is an automated way to modify structural shape based on predefined shape variables to find the optimal shape with the aid of a morph toolbox. The sensitivities of the response value to the design variables are solved with the method of feasible direction (MFD). MFD is a gradient-based algorithm, which is effective when the sensitivities of the system response, with respect to the design variables, can be computed easily and inexpensively; therefore, it is convenient for understanding the effect of the design variables and will most likely find the optima. Figure 3 presents the finite element model of the primary mirror assembly. 

### 3.3. Optimal Design Formulation

The goal of the optimization of the mirror component is to obtain the design parameter set that will minimize the mirror surface shape error (*RMS_T_*) caused by 30 °C uniform temperature decrease, while meeting the constraint requirements in Table 1. The mirror surface shape error (*RMS_T_*) caused by 30 °C uniform temperature decrease is set to be the objective function with the purpose of characterizing the thermal adaptability of the primary mirror assembly. The initial working temperature of the primary mirror assembly is set to normal room temperature 20 °C. The reason is that it is convenient for optical parts and structural parts processing and optical assembly and alignment under room temperature conditions. The optimization formulation of the primary mirror component can be defined in the following expressions. 

The objective function and the constraints are shown in Equation (4): (4)$$\begin{array}{ll} Minimize & f(X),\;f(X)=RMS_{T},\\ find & X={\left(d_{i},v_{j}\right)}^{T}\;for\;i=1,2,3,\ldots ,9;j=1,2,\ldots ,6.\\ S.T. & M\leq M_{0}\\ & RMS_{GR}\leq \delta_{0},\;RMS_{GZ}\leq \delta_{0}\\ & f_{1}\geq f_{0}\\ & d_{ilow}\leq d_{i}\leq d_{iup},\\ & v_{jlow}\leq v_{j}\leq v_{jup}. \end{array}$$
where *d_i_* is the thickness of the *i*-th mirror rib or faceplate and *v_j_* denotes dimensions related to compliance of the flexure. *M* is the current mass of the primary mirror component and *M*_0_ (*M*_0_ ≤ 1.6 kg) is the upper limit of the mass allocated by total performance index. *RMS_GR_*, *RMS_GZ_* are the values of the mirror surface shape error caused by gravity parallel and vertical to the optical axis, respectively. *δ*_0_ (*δ*_0_ ≤ 8 nm) is the upper limit of the mirror shape error. *f*_1_ is the current fundamental frequency, and *f*_0_ (*f*_0_ ≥ 200 HZ) is the lower limit of the fundamental frequency. *d_ilow_* and *d_iup_* are the lower limit and upper limit of *d_i_*. *v_jlow_* and *v_jup_* are the lower limit and upper limit of *v_j_*. 

In finite element based structural optimization, the structural fundamental formulas for linear static analysis is defined by Equation (5): (5)Ku=F
where *K* is the stiffness matrix and *u* is the displacement vector to be determined, and *F* is the applied force vector including nodal force, gravity, thermal load, etc. Differentiating this with respect to the design variable vector *X* (*X* = (*d_i_*, *v_j_*)*^T^*),
(6)$$\frac{\partial K}{\partial X}u+K\frac{\partial u}{\partial X}\;=\frac{\partial F}{\partial X}$$

The sensitivity of the displacement vector u with respect to the design variable vector *X* can be calculated as follows:(7)$$\frac{\partial u}{\partial X}=K^{-1}(\frac{\partial F}{\partial X}-\frac{\partial K}{\partial X}u)$$

The response vector *G* (*G* = (*RMS_T_*, *RMS_GX_*, *RMS_GY_*, *RMS_GZ_*, *M*)*^T^*) calculated from the displacements vector is written as:(8)$$G=P^{T}u$$
where *P^T^* is the corresponding coefficient matrix with respect to the *i*-th response function.

The sensitivities of the response vector *G* with respect to the design variable vector *X* can be expressed by Equation (9):(9)$$\frac{\partial G}{\partial X}=\frac{\partial P^{T}}{\partial X}u+P^{T}\frac{\partial u}{\partial X}$$

### 3.4. Results and Discussion 

In order to achieve optimal optical performance, the structure of the primary mirror and the flexures are parameterized together. The parameterized optimization model includes nine design variables for the primary mirror and six variables for the flexure. *d*_1_–*d*_6_ respect the thickness of the mirror ribs, and d7 denotes the thickness of the mirror faceplate. *v*_1_–*v*_6_ signifies dimensions related to compliance of the flexure. These parameters can be adjusted during the design process. The thickness of the mirror faceplate, the reinforcing ribs and the flexible leaf hinge involves mechanical processing and manufacturing limit, and a certain safety margin should be left. Consequently, the minimum thickness of the mirror ribs is 3 mm, and the faceplate is 4 mm separately for the initial lightweight design. Figure 4 depicts the parameterized optimization model of the primary mirror and the flexure. 

The objective function and the constraints settings can refer to Equation (4). The optimization problem is solved by means of Optistruct solver, in which, *RMS_T_*, *RMS_GR_*, *RMS_GZ_*, *M* and *f*_1_ are treated as the response function. Table 3 summarizes the lower bound, upper bound, initial and optimization results of the design variables. The final optimized primary mirror assembly can be achieved by updating the corresponding geometry dimensions. 

After multiple iterations, the optimized value of *d*_3_, *d*_4_, *d*_5_, *d*_6_ and *d*_8_ reached their respective lower bounds, whereas *d*_1_, *d*_9_ and *v*_5_ reached their respective upper bounds. *d*_1_ represents the thickness of the supporting hole rib of the primary mirror obtained the upper bound so that it has sufficient stiffness to resist gravity deformation. The thickness of the mirror faceplate *d*_7_ increased from the initial value of 4 mm to the optimized value of 6.6 mm. In addition, the size *d*_2_, *v*_3_ and *v*_4_ increased, on the contrary, *v*_1_ and *v*_6_ decreased to varying degrees. 

#### 3.4.1. Analysis of Wavefront Aberration Characteristics

The Fringe Zernike polynomials is a reordered subset of the Standard Zernike terms, which are applied to fit the deformed mirror surface. The 37 Zernike coefficients (removing the piston and tilt) under 30 °C uniform temperature decrease before and after the optimization were obtained as shown in Figure 5. The result indicated that the low-order aberration power (term 4) was the dominating part of thermal deformation, which significantly diminished after optimization and can be corrected by realignment and refocusing. Meanwhile, the second trefoil aberration B (term 20) is the secondary largest aberration which is of high-order aberration, and it is difficult to be eliminated by realignment and refocusing. In addition, other aberrations are reduced correspondingly after optimization.

#### 3.4.2. Sensitivity Analysis of Size Parameters

The method of feasible direction is adopted to solve the sensitivity of responses value to the design variables as mentioned in Section 3.2. The sensitivity of the response function with respect to the design variable can be expressed by Equation (9). Figure 6 shows the normalized percentage contributions of the size parameters to the response functions in four cases. From Figure 5a, the mirror size *d*_1_, *d*_5_, *d*_7_, and the flexure size *d*_8_ have significant influence on the mirror surface shape error (*RMS_T_*) caused by 30 °C uniform temperature decrease. From Figure 6b, the mirror size *d*_2_, *d*_5_ and the flexure size *d*_9_, *v*_4_ and *v*_6_ exert greater impact on the mirror surface shape error (*RMS_GR_*) caused by gravity vertical to the optical axis. From Figure 6c, the flexure size *d*_8_, *v*_3_ and *v*_4_ have obvious contribution to the fundamental frequency. From Figure 6d, the mass of the primary mirror component mainly depends on the mirror size *d*_1_, *d*_5_ and *d*_7_. The flexure size *v*_1_ and *v*_2_ have relatively weak effects on the four response functions above. In summary, each size exerts a certain impact on the performance in different cases. The sensitivity analysis is very useful for understanding which factors are the key, which provides a reference for the structural optimization design of aerial optoelectronic sensors. 

#### 3.4.3. Performance Evaluations and Comparisons

After multiple iterations, the optimization response function under gravity parallel and vertical to the optical axis and thermal load converged step by step as a whole. The iteration history of the mirror surface shape error in three cases is plotted in Figure 7. The initial and final values of the system performance responses are summarized in Table 4. The mirror surface shape error *RMS_T_* (removing the piston, tilt and power terms) caused by 30 °C uniform temperature decrease drastically diminishes from 26.5 nm to a minimum value 11.6 nm (less than *λ*/40), which is better than the performance requirements in Table 1. Namely, thermal deformation resistance of the primary mirror assembly has been obviously improved. Figure 8 depicts the primary mirror surface error map under 30 °C uniform temperature decrease before and after optimization. The mirror surface shape error *RMS_GR_* under gravity vertical to the optical axis rises from 6.3 nm to 8.0 nm, which still meets performance requirements as mentioned in Table 1. Furthermore, the mirror surface shape error *RMS_GZ_* under gravity parallel to the optical axis increases from 1.3 nm to 1.4 nm, which is far less than the design requirements. Moreover, the fundamental frequency of the primary mirror assembly decreases from 594.1 HZ to 203.1 HZ so that dynamic stiffness is sacrificed to a certain extent. However, it also satisfies the dynamic stiffness requirements. Furthermore, the mass of the primary mirror assembly increased from 1.3 kg to 1.6 kg mainly depends on the increase in thickness of the primary mirror faceplate which is still acceptable. 

There are various ways to evaluate the optical performance of the system. MTF (modulus of the OTF) is employed in this paper to assess the optical performance of the camera after the Zernike coefficients of the deformed surface are directly imported into the ray-tracing software Zemax [25]. The MTF values described below are the average value in the tangential and sagittal planes [26,27,28]. The image size of the visible light detector is 1920 × 1080 and the pixel size is 5.5 um corresponding to the cutoff frequency of MTF is about 90 cycles per mm. The field of view (FOV) the camera is 0.6° (azimuth) × 0.34° (pitch). As shown in Figure 9, in wavelength bandwidth of 450~700 nm, the value of MTF (0° field angle) under 30 °C uniform temperature decrease has declined significantly from 0.47 to 0.15 (before optimization) compared with optical design result (regardless of structural distortion) even though Zernike terms of the piston, tilt and power are subtracted. Compared to the initial design, the value of MTF (0° field angle) is improved from 0.15 to 0.21. The values of MTF (±0.24°, ±0.34° field angle, normalized ±0.7, ±1 field points) are inordinately improved after optimization as listed in Table 4. In other words, the optical performance of the camera under thermal load has been enhanced after optimization and thermal adaptability of the primary mirror has been obviously reinforced. The values of MTF (0° field angle) in the case of gravity parallel and vertical to the optical axis have slightly declined.

## 4. Experimental Verification

Based on the above optimized design results, the prototype of the primary mirror assembly was manufactured and assembled as shown in Figure 10. The precision of the structure should be guaranteed, especially the planarity of the mounting surface is suggested to be less than 3 nm. The invar sleeves were separately bonded into the support holes of the SiC primary mirror with the help of positioning tooling using epoxy adhesive (GHJ-01Z), and the flexures were also respectively fastened on the sleeves with bolts. High and low temperature cycle tests and a vibration test was used to release the stress of the primary mirror component during machining, assembly and gluing. 

Interferometry was performed on the primary mirror to acquire high measurement precision with a ZYGO interferometer working at 632.8 nm wavelength. Zernike terms of piston, tilt, power and coma which could be adjusted by alignment between mirror and interferometer were removed. An optical compensator containing two spherical lenses was adopted to correct the wavefront for measuring the aspheric surface of the primary mirror. The photograph of the interferometry scenario of the SiC primary mirror with the optical axis horizontal is displayed in Figure 11. The primary mirror surface shape error RMS after processing and coating is less than 0.025 *λ* at room temperature of 23 °C, which meets the performance requirement. The interferometry test result is shown in Figure 12. 

When subjected to a wide range of ambient temperature changes, the imaging quality of the high-resolution optical system will degrade differently. A ground temperature experiment was conducted to verify difference in imaging quality. The environmental temperature change simulation was carried out in a high- and low-temperature test chamber. A domestically produced target which was engraved with black and white strips was used to identify the resolution of the camera. Figure 13a shows the imaging photos taken by the visible camera as mentioned in Section 3.1 at room temperature of 20 °C. It is not difficult to identify that the minimum distinguishable stripe is the 23rd group. Then the camera was put into a high and low temperature test chamber and the internal temperature is set to −10 °C without thermal control facilities. After being placed for two hours, the internal temperature environment reached a state of thermal equilibrium. The target image obtained by the visible system is shown in Figure 13b. It can be seen that the smallest discernible stripe is the 16th group under low temperature condition. The result demonstrates that although the image quality of the camera degrades, it still meet the requirements of the index.

## 5. Conclusions

The purpose of this article is to further optimize the thermal adaptability of the supporting structure of the primary mirror. Integrated optomechanical analysis and optimization for the primary mirror assembly are put forward. The mirror surface shape error caused by uniform temperature decrease is treated as the objective function, and the fundamental frequency of the mirror assembly and the surface shape error caused by gravity parallel or vertical to the optical axis are taken as the constraints. A detailed size optimization is conducted to optimize its size parameters. The contribution of size parameters to the surface shape error, fundamental frequency and mass are further evaluated. The configuration of the primary mirror and the flexure are acquired. The results demonstrate that the size parameters significantly have an effect on the optical performance and which dimension parameters are of the critical factors. The mirror surface shape error caused by 30 °C uniform temperature decrease effectively decreased from 26.5 nm to 11.6 nm, although the weight of the primary mirror assembly increases by 0.3 kg. The mirror surface shape error under gravity parallel and vertical to the optical axis grows slightly. The fundamental frequency of the primary mirror assembly decreases from 594.1 HZ to 203.1 HZ, but still satisfies the dynamic stiffness requirements. The value of the system’s MTF (0° field angle) is improved from 0.15 to 0.21 after optimization. The values of MTF (±0.24°, ±0.34° field angle, normalized ±0.7, ±1 field points) are inordinately improved after optimization. In other words, the optical performance of the camera under thermal load has been enhanced and thermal adaptability of the primary mirror has obviously been reinforced. Finally, a prototype of the primary mirror assembly is manufactured and assembled. A ground thermal test was developed to verify variation of imaging quality at room and low temperature, respectively. The image quality of the camera meets the requirements of the index despite degradation. This research provides a meaningful reference for the structural optimization design of aerial optoelectronic sensors. 

## Figures and Tables

**Figure 1 sensors-21-07993-f001:**
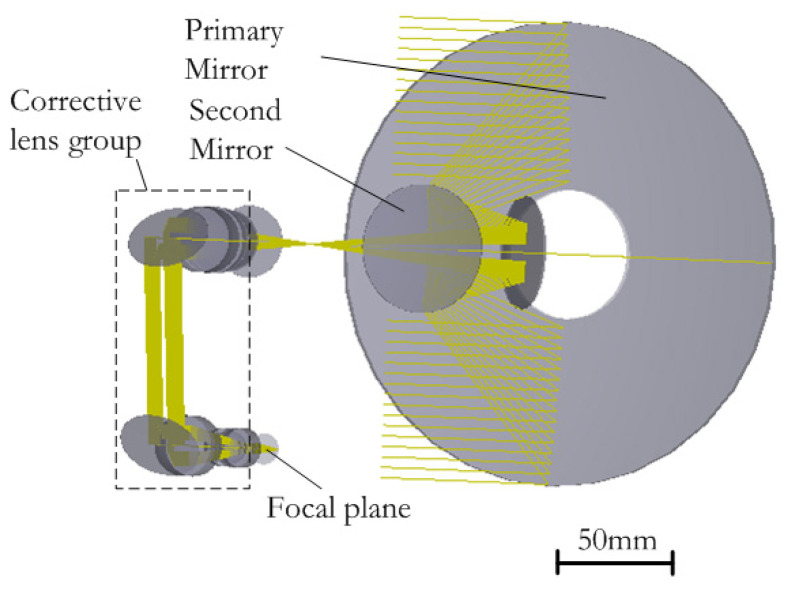
Optical model of the visible light camera.

**Figure 2 sensors-21-07993-f002:**
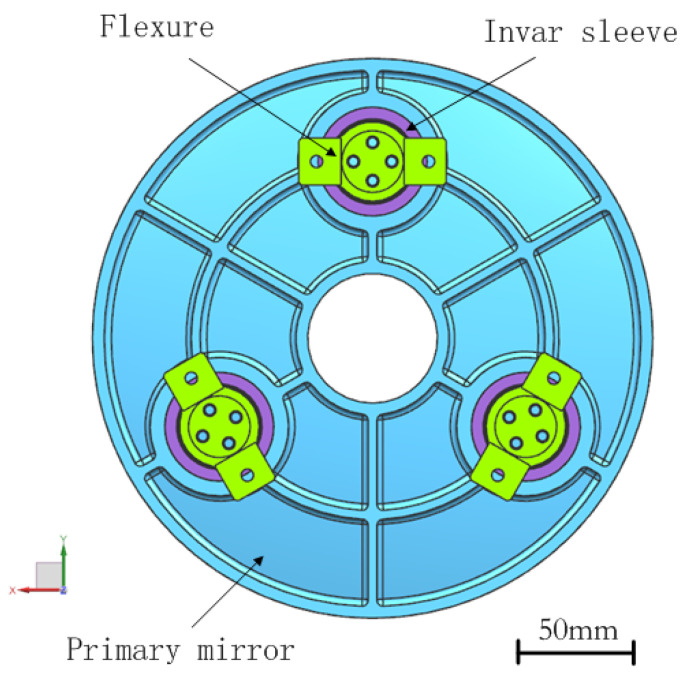
Schematic of the primary mirror assembly.

**Figure 3 sensors-21-07993-f003:**
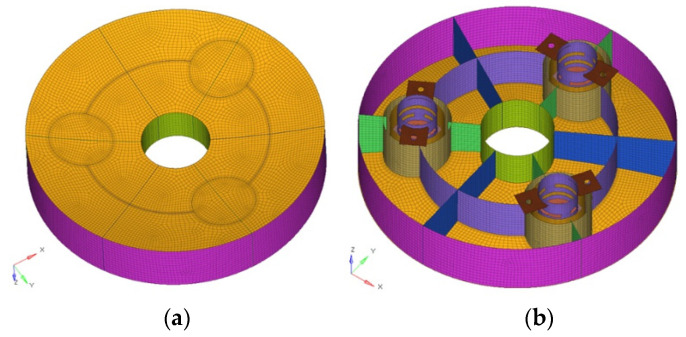
Finite element model of the primary mirror assembly. (**a**) Front view. (**b**) Back view.

**Figure 4 sensors-21-07993-f004:**
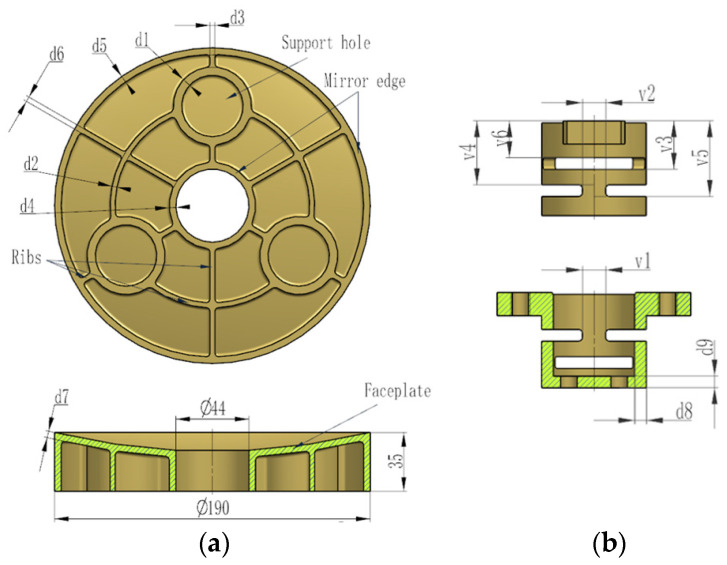
Design variables for size optimization. (**a**) Primary mirror. (**b**) Flexure.

**Figure 5 sensors-21-07993-f005:**
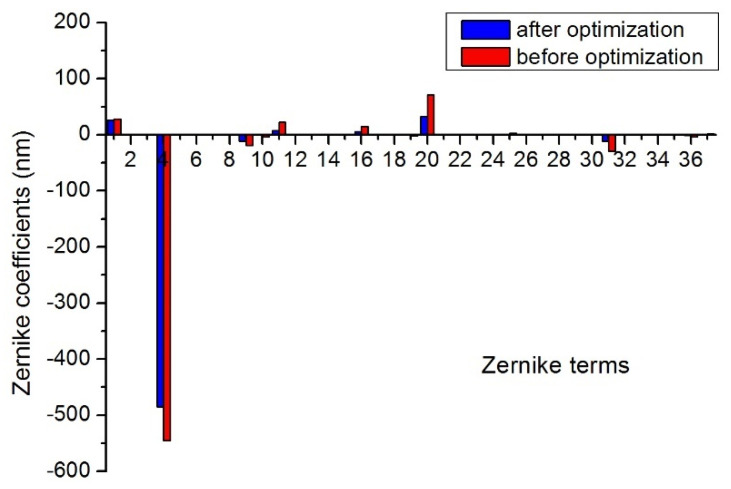
Zernike coefficients under 30 °C uniform temperature decrease before and after the optimization.

**Figure 6 sensors-21-07993-f006:**
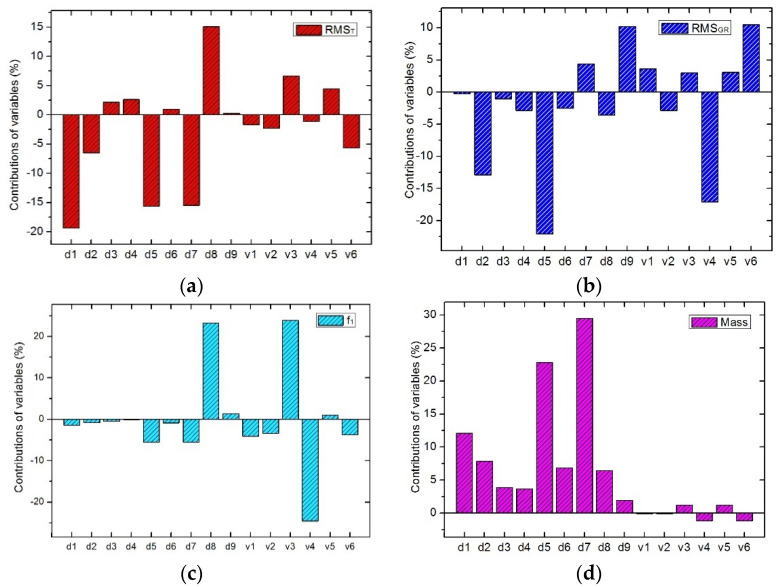
Sensitivity of size parameters. (**a**) The mirror surface shape error RMS_T_. (**b**) The mirror surface shape error RMS_GR_. (**c**) Fundamental frequency. (**d**) Mass of the primary component.

**Figure 7 sensors-21-07993-f007:**
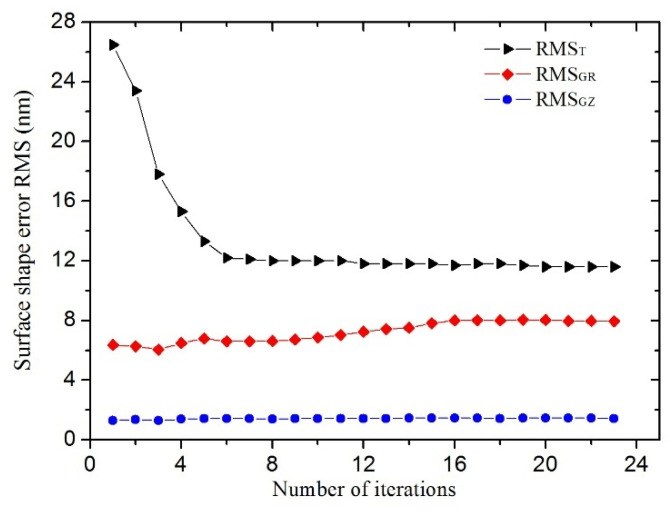
Iterative history of the mirror surface shape error.

**Figure 8 sensors-21-07993-f008:**
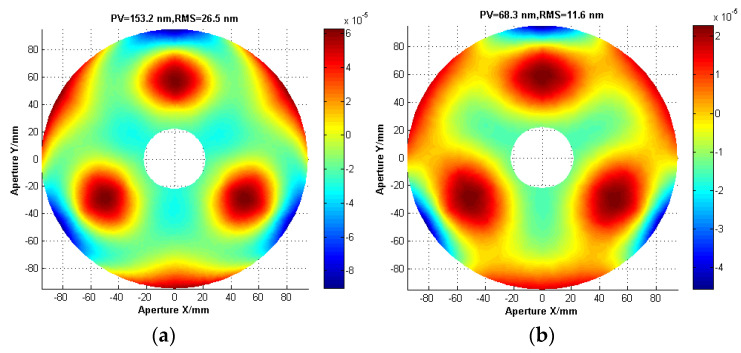
Surface error map under 30 °C uniform temperature decrease. (**a**) Initial design. (**b**) Optimized design.

**Figure 9 sensors-21-07993-f009:**
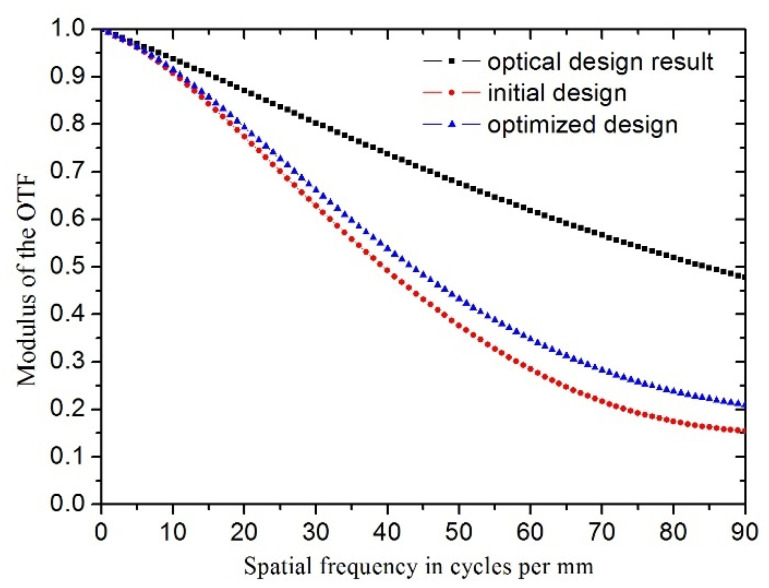
Comparison of the modulus of the OTF (MTF, 0° field angle) curves of initial and optimized design under 30 °C uniform temperature decrease.

**Figure 10 sensors-21-07993-f010:**
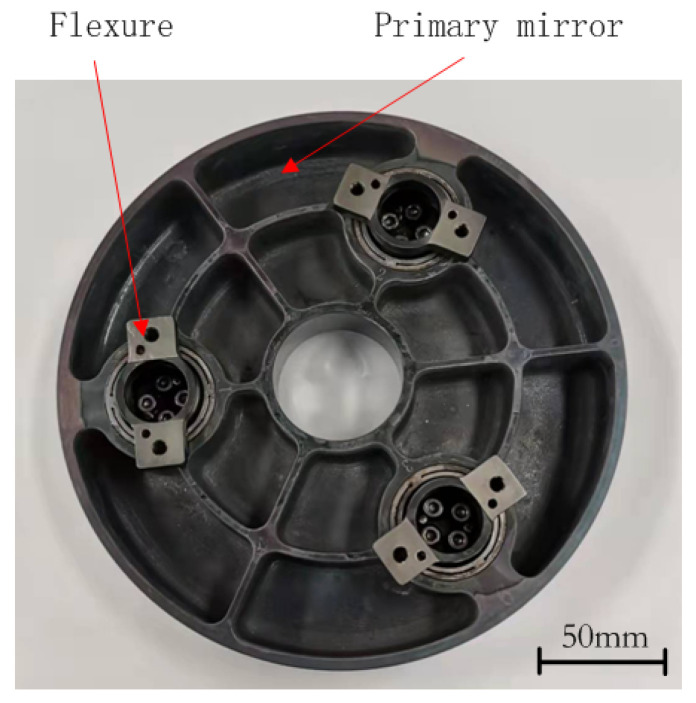
Prototype of the primary mirror assembly.

**Figure 11 sensors-21-07993-f011:**
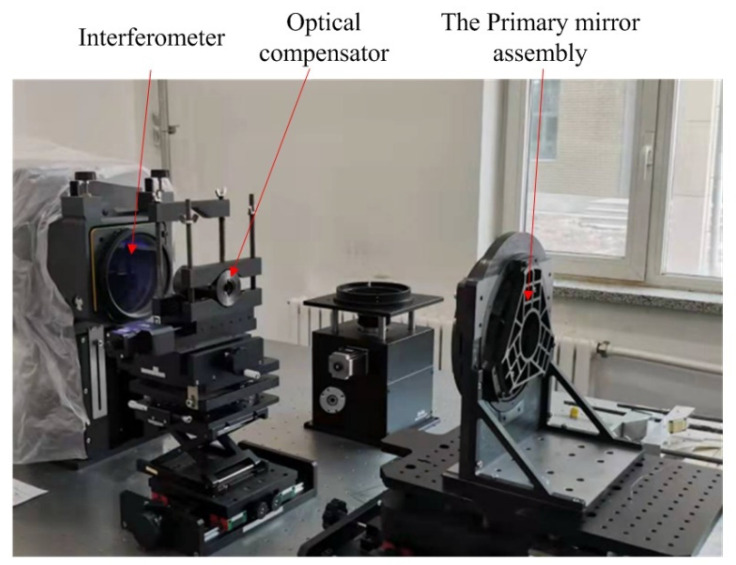
Interferometry scenario of the SiC primary mirror with the optical axis horizontal.

**Figure 12 sensors-21-07993-f012:**
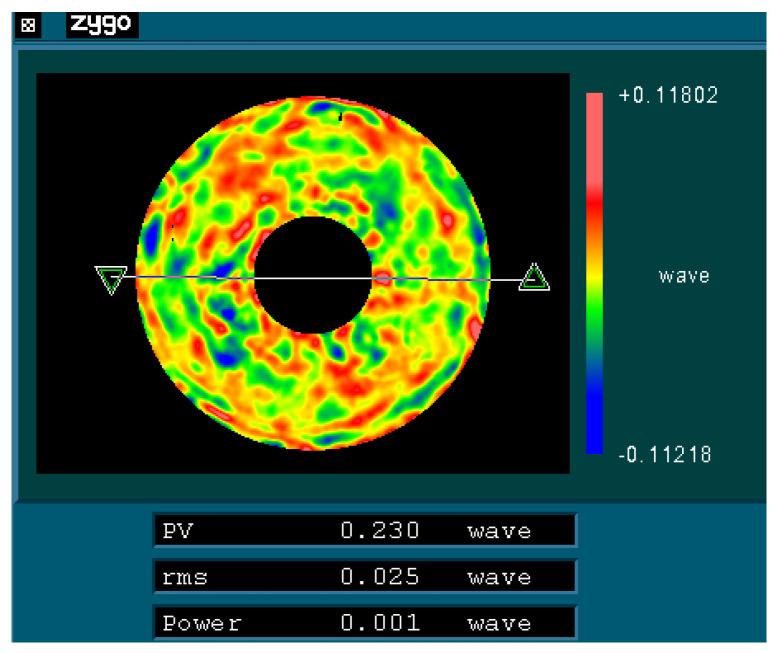
Interferometry test result of the primary mirror.

**Figure 13 sensors-21-07993-f013:**
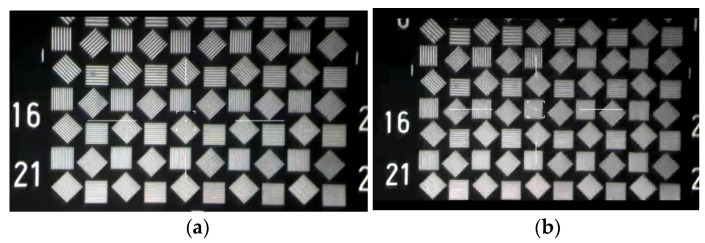
Target images acquired at different temperatures. (**a**) At the temperature of 20 °C. (**b**) At the temperature of −10 °C.

**Table 1 sensors-21-07993-t001:** Performance requirements of the primary mirror assembly.

Total Mass	≤1.6 kg
RMS error under 1G gravity parallel or vertical to the optical axis	≤8 nm
RMS error under uniform temperature decrease 30 °C	≤15.6 nm
Fundamental frequency	≥200 HZ
Operating temperature	20 ± 30 °C
The primary mirror’s tilt under 1G gravity vertical to the optical axis	≤20″
The primary mirror’s decenter under 1G gravity vertical to the optical axis	≤20 um

**Table 2 sensors-21-07993-t002:** Material properties of the mirror assembly.

Components	Material	Young’ Modulus (Gpa)	Density (g/cm^3^)	Coefficient of Thermal Expansion (10^−6^/K)	Poisson’ Ratio
Primary mirror	SiC	350	3.0	2.5	0.25
Invar sleeve	4J36	143	8.12	2.43	0.25
Flexure	TC4	107	4.44	8.9	0.34

**Table 3 sensors-21-07993-t003:** Initial and optimized values of the design variables.

Design Variable	Lower Bound (mm)	Upper Bound (mm)	Initial Value (mm)	Optimized Value (mm)
*d* _1_	4.0	8.0	4.0	8.0
*d* _2_	3.0	6.0	3.0	5.4
*d* _3_	3.0	6.0	3.0	3.0
*d* _4_	3.0	6.0	3.0	3.0
*d* _5_	3.0	6.0	3.0	3.0
*d* _6_	3.0	6.0	3.0	3.0
*d* _7_	4.0	8.0	4.0	6.6
*d* _8_	2.0	6.0	2.0	2.0
*d* _9_	3.0	6.0	3.0	6.0
*v* _1_	1.1	7.6	6.0	5.8
*v* _2_	1.1	7.6	6.0	6.0
*v* _3_	9.5	15.5	12.5	15.1
*v* _4_	13.5	19.5	16.5	16.7
*v* _5_	16.5	22.5	19.5	22.5
*v* _6_	6.5	12.5	9.5	7.7

**Table 4 sensors-21-07993-t004:** Comparison of initial and optimal primary mirror component design.

Terms	Initial Value	Optimized Value
*RMS_T_* (nm)	26.5	11.6
*RMS_GR_* (nm)	6.3	8.0
*RMS_GZ_* (nm)	1.3	1.4
*f*_1_ (HZ)	594.1	203.1
Mass (kg)	1.28	1.59
Lightweight ratio (%)	45.8	33.3
*MTF_T_* (0° field angle)	0.15	0.21
*MTF_T_* (+0.24° field angle)	0.13	0.17
*MTF_T_* (−0.24° field angle)	0.14	0.16
*MTF_T_* (+0.34° field angle)	0.09	0.10
*MTF_T_* (−0.34° field angle)	0.10	0.11
*MTF_GR_* (0° field angle)	0.46	0.43
*MTF_GZ_* (0° field angle)	0.47	0.46

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
