# Peer review of "Design and Optimization for Mounting Primary Mirror with Reduced Sensitivity to Temperature Change in an Aerial Optoelectronic Sensor"

_sensors, 2021, doi:10.3390/s21237993_

Round 1
Reviewer 1 Report
Dear Editor,
I have read the manuscript sensors-1454037 entitled: “Design and optimization for mounting primary mirror with reduced sensitivity to temperature change in an aerial optoelectronic sensor” and I would like to address following suggestions to the authors:
At line 18: primary mirror - should be - primary mirrors
At line 31: prototype of the - should be – a prototype of the
At line 44: of high resolution - should be –of a high resolution
At line 45: environmental factor, - should be – environmental factors
At line 72: lightweight mirror - should be – lightweight mirrors
At line 80: optimization of mirror - should be – optimization of the mirror
At line 84: mirror of aerial - should be – mirror of an aerial
At line 101: optical model - should be – the optical model
At line 137: mirror assembly - should be – mirror assemblies
At line 144: extraordinary suitable - should be – extraordinarily suitable
At line 147: holes is adopted - should be – holes are adopted
At line 148: improving lightweight - should be – improving the lightweight
At line 192: of morph toolbox. - should be – of a morph toolbox.
At line 212: M is current mass of primary - should be – M is the current mass of the primary
At line 216: is lower limit - should be – is the lower limit
At line 239: v1-v6 signify dimensions - should be – v1-v6 signifies dimensions
At line 284: weak effect - should be – weak effects
At line 284: response function above. - should be – response functions above.
At line 287: optoelectronic sensor. - should be – optoelectronic sensors.
At line 297: final valures - should be – final values
At line 338: suggested to less than - should be - suggested to be less than
At line 348: two spherical lens was - should be - two spherical lenses was
At line 349: measuring aspheric - should be - measuring the aspheric
At line 349: of interferometry - should be - of the interferometry
At line 372: different temperature.- should be – different temperatures.
At line 392: Finally, prototype - should be - Finally, a prototype
At line 396: optoelectronic sensor. - should be - optoelectronic sensors.
Author Response
Dear reviewer,
Thank you so much for pointing out our numerous errors of grammar so carefully. Please see the attachment.
Thanks for your time.
Sincerely,
Meijun Zhang on behalf of all the authors

Reviewer 2 Report
In this paper, the integrated optomechanical analysis and optimization are proposed for the primary mirror assembly of the aerial optoelectronic sensor. In the optimization process, shape error of the mirror is set as the objective function. The results demonstrate that the size parameters have significant effect on the optical performance. The fundamental frequency of the primary mirror decreases from 594.1 Hz to 203.1 Hz. Both the optical performance and thermal adaptability of the camera have been enhanced after optimization. However, there are some questions:
For the optical mechanical system, its performance depends not only on the performance of its primary mirror under various conditions, but also on other components such as secondary mirror and so on. This paper mainly optimizes the main mirror assembly. To better show the thermal adaptability of the whole system under different temperatures? can the system be optimized as a whole?
In this paper, the imaging performance of the optimized system is represented by MTF with 0 field angle. When the optical system is used in a aerial optoelectronic sensor camera, can you specify the FOV of the camera? And how does the optimized system perform in other FOVs?
“30oC uniform temperature decrease” is frequently mentioned in this paper. How is it embodied in the optimization? What is the temperature range it refers to?
Author Response
Dear reviewer,
Please see the attachment.
Thanks for your time.
Sincerely,
Meijun Zhang on behalf of all the authors.

Reviewer 3 Report
This manuscript combined and studied together the optimization design of the mirror and its supporting structure. High thermal deformation resistance of the primary mirror of aerial optoelectronic sensor is considered for better adaptation in the low or high temperature working environment. A detailed size optimization of the primary mirror and the flexure is conducted together to optimize their dimension parameters. It is practical. However, I think this manuscript is an application-oriented paper. There are not experimental or theoretical novelty when compared to existing literature. Its contribution is not enough. In the optimization processing, multi-merits function and boundary limits are necessary and basic to perform this kind of work. The authors must clarify which contributions were done in this paper and compared with other optimization processing, not limited in the specific mounting primary mirror.
Author Response

(The authors gave the same response as above.)

Reviewer 4 Report
This review covers design and optimization for imaging optics with reduced sensitivity to temperature change. This topic is of particularly high technical relevance for precision opto-mechanis. Theoretical modeling and optimization approaches (material analysis and mechanics of materials) are well organized and presented. In addition, experimental data verified the optimization results well. It will be helpful for researchers interested in optimization for optics design in large temperature change environment. There are several issues to be corrected and the paper needs improvement before being publishable.
For example,
- Overall, the analysis is done with a steady state condition. Can you provide any analysis or discussion on transient conditions?
- A scale bar in the figures would be helpful to confirm the size of parts.
- Line 179: fem ->FEM
- Line 196: the optima [].??
- Line 261: In, Figure 5 Zernike coefficients are given, however please provide the definition of Zernike polynomials and its coefficients
- Line 277: Figure 5 (6), ??
- Line 367: “… obtained by the visible system is shown in Figure 13 (b). It can be seen that the smallest discernible stripe is the 16th group under low temperature condition. The result demonstrates that although the image quality of the camera degrades, it still meet the requirements of the index.” -> Could you specify the difference and similarity in Fig 13 based on the analysis? It is difficult to pick pixels to be compared in Fig. 13.
Author Response

(The authors gave the same response as above.)

Round 2
Reviewer 3 Report
Now, it can be accepted due to the well-written version.